# Mental Health and Wellbeing in Young People in the UK during Lockdown (COVID-19)

**DOI:** 10.3390/ijerph19031132

**Published:** 2022-01-20

**Authors:** Matthew Owens, Ellen Townsend, Eleanor Hall, Tanisha Bhatia, Rosie Fitzgibbon, Francesca Miller-Lakin

**Affiliations:** 1Department of Psychology, University of Exeter, Exeter EX4 4QG, UK; elmh202@exeter.ac.uk (E.H.); tb542@exeter.ac.uk (T.B.); rf395@exeter.ac.uk (R.F.); fm407@exeter.ac.uk (F.M.-L.); 2Self-Harm Research Group, School of Psychology, University of Nottingham, Nottingham NG7 2RD, UK; Ellen.Townsend@nottingham.ac.uk

**Keywords:** wellbeing, depression, stress, sleep, rumination, young adults, COVID-19

## Abstract

This study aimed to assess the levels of mental wellbeing and potential for clinical need in a sample of UK university students aged 18–25 during the COVID-19 pandemic. We also tested the dose-response relationship between the severity of lockdown restrictions and mental wellbeing. We carried out a prospective shortitudinal study (one month between baseline and follow up) during the pandemic to do this and included 389 young people. We measured a range of facets of mental wellbeing, including depression, depressogenic cognition (rumination), wellbeing, stress and sleep disturbance. Our primary outcome was ‘probable depression’ as indexed by a score of ≥10 on the patient health questionnaire (PHQ-8). The prevalence of probable depression was significantly higher than pre-pandemic levels (55%) and did not decrease significantly over time (52%). Higher levels of lockdown severity were prospectively associated with higher levels of depressive symptoms. Nearly all students had at least one mental wellbeing concern at either time point (97%). The evidence suggests that lockdown has caused a wellbeing crisis in young people. The associated long-term mental, social, educational, personal and societal costs are as yet unknown but should be tracked using further longitudinal studies.

## 1. Introduction

As a way of attempting to control the spread of the SARS-CoV-2 virus and slowing COVID-19 transmission, many countries around the globe implemented non-pharmaceutical interventions (NPI) in an effort to limit social contact throughout society. Collectively these NPIs have become known in some quarters as lockdown. In the case of the UK, a number of strategies to reduce the spread of infection have thus far been tried, including: limiting non-essential contact and travel, stay at home orders, social distancing, home-working, school closure, limiting outdoor social gatherings (the ‘rule of six’), curfews for hospitality and remote online learning. The first national lockdown in the UK occurred on the 23rd of March 2020 and saw schools closed to most pupils and the end of face-to-face teaching in universities. The latter was replaced with digital delivery of synchronous and asynchronous online lectures and seminars. The UK government perceived the use of national lockdowns as necessary, primarily to control the virus, protect the National Health Service (NHS) and save lives. However, such strategies may have unintended negative consequences for all strata of society.

As has been pointed out elsewhere, societal lockdowns may inadvertently lead to poor outcomes, such as unemployment, increased child abuse and domestic violence, and reduced non-COVID-19 health care, such as cancer screenings and vaccinations [1]. For example, a recent analysis suggests that from April to October 2020 there were more than 3500 fewer people diagnosed and treated for colorectal cancer in England than would have been expected [2]. In this way lockdowns may paradoxically lead to an increase in excess all-cause mortality [1]. In addition, lockdowns may increase substance abuse and numbers of suicides [3,4], increase levels of isolation in older adults [5] and elevate levels of psychological distress [6].

There is now gathering evidence that a number of different facets of mental wellbeing have suffered during the time of lockdowns across the globe, including but not limited to anxiety, depression, worry and rumination, stress, wellbeing and sleep [7,8,9,10,11,12,13]. For example, early on in the course of the pandemic (May 2020), initial studies in the U.S. found psychological distress levels for the previous month as high as those estimated for the previous year [14]. Similarly in Germany, at the same time, a study found that 31% met the criterion for probable depression on the patient health questionnaire (PHQ) [15]. In an early study carried out in the Basque Autonomous Community, young people (18–25) were also disproportionately affected by stress, anxiety and depression [16]. It was also shown in a large UK cohort that mental wellbeing decreased after the first month of lockdown, particularly for 18–24 year olds [17].

The adolescent to young adult age range (14–24) is a complex sensitive period for the emergence of mental health difficulties [18,19]. Indeed, approximately half of mental health disorders emerge in the teenage years [20] and three quarters emerge by 24 years of age [21]. The sequalae of poor mental health and low wellbeing are numerous and include medium and longer-term negative effects, ultimately preventing individuals from reaching their full potential [22]. Common mental health difficulties, like depression, are associated with significant negative outcomes, including low educational attainment [23], relationship disruption, unstable employment and an increased risk of mortality through somatic illness and suicide [24]. Recently in the UK, it has been shown that the economic disruption for young people (18–24) following the response to the pandemic led to feelings of defeat, entrapment, shame and hopelessness. It was found that by January 2021, 20% of those young people who were in employment before the pandemic were no longer working. Furthermore, the report demonstrated a prospective link between experiencing economic disruption and suicidal thoughts. Young people experiencing economic disruption were almost one and a half times more likely to experience suicidal thoughts three months later [25].

Unfortunately, individuals suffering with common mental health difficulties, like depression, can also expect poor long-term economic outcomes, with personal earnings considerably lower than those without a mental disorder [26]. In the short-term, students with mental health problems are at increased risk of dropping out of university. In the UK, for example, the rate of drop out for students experiencing mental health problems has increased over 200% from 2009 to 2015 [27].

Given this developmentally sensitive period, highlighting a population vulnerable to mental health problems, it is perhaps unsurprising that the policy response to the COVID-19 pandemic would give rise to further mental health problems in young people, as it has in previous pandemics [28,29]. Importantly, although young people are relatively spared from the physical effects of COVID-19 [30,31,32] and not significant drivers of transmission [33], they are most vulnerable to suffering low mental wellbeing as a result of this pandemic [34]. Assessing the success or otherwise of lockdown policies in this way is essentially adopting a cost-benefit analytic approach. In the present study we set out to assess the level of mental wellbeing harms and potential for clinical need in young people during the COVID-19 pandemic.

One approach to estimating the magnitude of potential for clinical need has been to use cut-off points on validated measures of mental wellbeing and compare them to pre-pandemic estimates. For instance, a recent meta-analysis of 12 studies including adult samples, estimated a depression prevalence rate of 25%, which is a seven-fold increase on pre-pandemic prevalence [35]. Consistent with other research [36], this meta-analysis suggested that young people may be particularly affected, including university students [37]. For example, a longitudinal study carried out at the early stages of the pandemic reported a significant rise in depressive symptoms linked to a reduction in wellbeing and sleep quality in university students [9]. The authors of this study found that 34% of the sample met the criteria for clinical depression in the Hospital Anxiety and Depression Scale (HADS).

There is, however, less information on the mental wellbeing of students further into the pandemic. Specifically, it is unclear whether the high prevalence of mental health problems seen early on has remained through the pandemic. It also remains unresolved whether the level of public health restrictions or lockdown has played a causal role in the lower mental wellbeing of young people. In the case of Italy there is evidence that, in their strict lockdown of spring 2020, a longer duration of isolation was related to worse mental wellbeing in adults [38]. However, the effects of the level of restriction on mental wellbeing were impossible to test given that the entire country was in the same level of restriction. In Germany, higher levels of lockdown restrictions were associated with more psychological distress and lower life satisfaction but not psychopathology per se [15]. A multi-country analysis (77 countries), however, tested the hypothesis that lockdown severity that restricted access to outside natural environments would be positively associated with poor mental health [39]. In this study, a three-level lockdown severity categorisation was used (1 = most severe, e.g., China, Italy; 2 = severe but with some access to outside space, e.g., UK, France; 3 = least severe, e.g., Sweden) to classify severity across countries, and the results showed that depressive symptoms were higher in countries with higher levels of restriction severity.

### The Current Study

Here we assessed the mental wellbeing of UK university students at two timepoints that were a month apart during the COVID-19 lockdown. Initial data collection using the survey (herein T1) began in December 2020 and was repeated (T2) in January 2021. Please see Figure 1 for a study timeline in the context of the UK lockdown. We measured the retrospective estimates of change in wellbeing and health behaviours of young people currently in lockdown, as well as their current mental wellbeing using validated psychometric instruments. We repeated the mental wellbeing assessment one month later as the various lockdown restrictions continued. Relatively short follow-up periods have been recommended in mental health research with young people, given that the predictive power of risk factors for mental health difficulties can weaken as follow-up length increases [40]. We tested the following a priori hypotheses:Depression, depressogenic cognition (rumination), wellbeing, stress and sleep disturbance (collectively: mental wellbeing) and diet and exercise (health behaviours) would be poorer for young people during the lockdown, relative to pre-pandemic levels.Mental wellbeing would be worse for those in higher relative to lower lockdown restrictions.The negative effects of lockdown on mental wellbeing would persist from T1 to T2.

## 2. Materials and Methods

This study was a shortitudinal; a repeated survey over a one-month period during the COVID-19 lockdown.

### 2.1. Sample Size Calculation

There were two considerations to our sample size calculations. First, we wanted to be able to detect a higher proportion of young people with probable depression on the PHQ-8, our primary outcome, than in studies carried out before the COVID-19 pandemic. To do this we note that Kroenke [41] found a 6.2% prevalence rate on this measure in 18–24-year-olds. Although studies suggest rates ~30% or higher, we conservatively estimated that we would find a prevalence rate of 15%. To have 90% power to detect this difference using a Wald test would require 173 participants. The second effect of interest was whether there would be any association between the severity of lockdown restrictions and depression scores. We used Monte Carlo Simulation (MCS) in Stata (version 16.0) to estimate the likely sample size required to detect a standardised regression coefficient of 0.2 using an SEM model representing the association between lockdown restriction and depression at T1. Using 2000 replications, it was estimated that a sample size of 400 would give 85% power to detect such an effect.

### 2.2. Participants

We recruited 389 young people (mean age = 21.04, sd = 1.62; range = 18–25; female = 299, 76.86%, male *n* = 81, 20.82%, other = 9, 2.31%) primarily studying at university in undergraduate (*n* = 335, 86.12%), postgraduate or medical programs (*n* = 41, 11.83%) or in other life circumstances (*n* = 8, 2.06%). The sample size at T1 had the effect of marginally reducing the power in the study from 85% to 84%. In the event, 254 participants completed the T2 assessment (65.30%), reducing the power at this time point to 65%.

### 2.3. Measures

#### 2.3.1. Retrospective Questions on Change

The retrospective questions on perception of change in mental wellbeing and health behaviours included stress, sleep, diet and physical activity and were based on those used in previous research [8]. Statements included were as follows: “How have your stress levels changed since the Covid-19 pandemic started?”, “How has your amount of sleep changed since the start of the pandemic?”, “How has your diet changed since the COVID-19 pandemic started?” and “How has your amount of physical activity changed since the COVID-19 pandemic started?” Responses were measured using a 5-point scale, ranging from “A lot worse” to “A lot better” [8]. Scores per item range from 1–5.

#### 2.3.2. Depression

Symptoms of depression was the primary outcome, measured using the Patient Health Questionnaire (PHQ-8), which is an 8-item version of the original scale which assesses the severity of depressive symptoms [42] but excludes the item on suicidality. Participants are asked to indicate how often during the past two weeks they were bothered by a number of symptoms, such as “Feeling tired or having little energy”. Responses are recorded on a four-point Likert scale (0 = “Not at all” to 4 = “Nearly every day”). Scores are summed (range = 0 to 24), with higher scores meaning higher symptom severity. The PHQ-8 has shown good reliability, sensitivity and specificity [41,43]. A score of ≥10 indicates at least moderate symptoms and is used as a cut-off for suggesting probable depression [41]. In the present study the measure demonstrated good reliability (T1 and T2 α = 0.87).

#### 2.3.3. Wellbeing

The Warwick Edinburgh Mental Wellbeing Scale (WEMWBS) is a well-validated measure of wellbeing, suitable for young people [44]. We used the short 7-item version in the present study [45]. The scale asks participants to reflect over the last two weeks (e.g., “I have been feeling optimistic about the future”) and responses are coded using a 5-point scale ranging from 1 (“None of the time”) to 5 (“All of the time”), scores range from 1 to 35. The WEMWBS has demonstrated good internal reliability in student samples [46] and test-retest reliability over one week is high [45]. An average score of 19.25 and 19.98 at the 15th percentile (approximately one SD below the mean) was reported in a large sample of 15,244 and 11,948 females and males, respectively [47]. Therefore, a conservative cut-off of 19 was used in the current study to indicate poor mental wellbeing. In the current study scale reliability was good (T1 *α* = 0.81; T2 *α* = 0.85).

#### 2.3.4. Sleep Disturbance

The four-item Jenkins Sleep Scale (JSS) is a brief standardised measure of sleep disturbance over the past month (e.g., “Waking up feeling tired”) rated, asking questions such as “How often in the last month did you have problems falling asleep?”. Participants are asked to respond using a 6-point Likert scale (0 = “Not at all” to 5 = “22–31 days”), giving summed scores that range from 0–20, with high scores indicating greater sleep disturbance. The scale is unidimensional and reliable [48] and a score of ≥12 is indicative of high frequency sleep disturbance [49]. In a community study of 2515 individuals, the unidimensionality of the JSS was replicated and it was shown that 6% had a score greater than 12 [50]. In the present sample internal reliability was good (T1 *α* = 0.79; T2 *α* = 0.78).

#### 2.3.5. Rumination

We used the five-item brooding subscale of the Ruminative Response Scale [51,52] to measure rumination. Participants are asked to respond to five statements (e.g., “What am I doing to deserve this”) using a five-point scale (1 “Almost never” to 4 “Almost always”). The scale has been shown to be reliable in previous research [53] and in the present sample (T1 *α* = 0.83; T2 *α* = 0.84). A score of 12.36 on the brooding scale represents one SD above the mean [52] and, based on this, we used a score of ≥ 13 to indicate high levels of rumination. The estimated prevalence of high levels of rumination is therefore 15%.

#### 2.3.6. Perceived Stress

The Perceived Stress Scale [54] is a widely used 4-item stress scale shown to have good internal consistency and reliability [55]. Respondents answered questions about thoughts and feelings in the past month, such as “How often have you felt you were unable to control the important things in your life?”. Participants indicated their response from 0 (“Never”) to 4 (“Very often”). PSS-4 scores were obtained by reverse coding the positive items (Items 2 and 3) and then summing all four items. Scores ranged from 0–16, with higher scores indicating more stress. We used scores of ≥ 8 to indicate high stress in line with previous research that showed that the majority (75%) of 7180 18-29 year-olds scored up to this level [55]. The scale reliability was also good in the present analysis (T1 *α* = 0.70; T2 *α* = 0.74).

#### 2.3.7. Lockdown Status

In the UK, a stepped approach was taken to restrict the movement of citizens that increased in severity to the maximum level. A national lockdown imposed the highest level of restriction to the whole nation. The four nations of the UK adopted similar approaches and in England a four-tiered model was used (Tiers 1–4), plus national lockdowns. During the current study the UK experienced it’s third lockdown on 6 January 2021. The first lockdown occurred on 23 of March 2020, and the second occurred on 5 of November 2020.

#### 2.3.8. Procedure

Participants were recruited via opportunity sampling. Eligible volunteers were found through student groups and social media. The study consisted of two online surveys which were compiled using the software Qualtrics. Participants read an information sheet and gave consent prior to starting the initial survey (T1). Participants were then presented with the study measures. It was made clear to all young people that participation in the study was entirely voluntary and that they could end the questionnaire at any point if they felt distressed or uncomfortable. Participants scoring 10 or greater on the PHQ-8 were immediately signposted to sources of mental health support. Approximately one month after the T1 survey, participants were sent the second and final survey (T2) by email. The researchers contacted participants up to three times before coding them as missing (dropout). Participants were given a written debrief form explaining the study in further detail at the end of the T2 questionnaire, or at the point they decided to leave the study. Participants were offered the opportunity to partake in a prize draw for a single £25 voucher each time they completed one of the two surveys.

#### 2.3.9. Statistical Analysis

We report the percentage of agreement with the retrospective questions and assessed gender differences using chi-square tests of association. The observed proportions of poor wellbeing (depression, wellbeing, sleep rumination and stress) in the current study were compared with the previous estimates ascertained from the literature using two-sample tests of proportions. To test for any association between lockdown restriction severity and wellbeing we used the structural equation modelling (SEM) command in STATA to jointly estimate the effect of lockdown restrictions on the five measures of wellbeing. For this analysis we categorised lockdown status into high severity (national lockdown and Tier 4) and low severity (Tiers 1–3). It is important to note that at T2, the UK was in a national lockdown, with all regions experiencing broadly the same level of restrictions. We conducted this analysis cross-sectionally at T1 and prospectively at T2. We adjusted for age and sex in each analysis. We checked for any associated between drop out from the study and baseline characteristics. We made an a priori decision to use full information maximum likelihood (FIML) estimation in the SEM models to account for any effects of missing data. FIML is a robust unbiased and efficient technique outperforming traditional approaches to missing data [56].

## 3. Results

### 3.1. Retrospective Views on Change through the Pandemic

Self-reports of intra-individual change comparing pre-pandemic to current status varied on all items and included negative, positive, as well as no change. However, the majority of the sample reported that stress had increased either a little or a lot (84.17%), that sleep had got a little or a lot worse (59.10%), that their dietary habits had got either a little or a lot more unhealthy (53.83%), and that they had engaged in either a little or a lot less physical activity (55.41%). The pattern in the data (see Figure 2) suggested that the changes may be moderated by sex but only one difference was statistically significant (females reported a large increase in stress relative to males (*x*^2^ = 10.35, df(4), *p* = 0.035)).

### 3.2. Proportion of Participants with Potential for Clinical Need

#### 3.2.1. Depression

The reference sample for rates of depression comes from Kroenke et al.’s (2009) review of the PHQ-8, in which 6.2% of 18–24-year-olds met criteria for probable depression. In the present sample 55.5% (T1) and 52.8% (T2) of young people had probable depression. A two-sample test of proportions showed that the large difference between 6.2% and the present estimates at both T1 and T2 was unlikely to be due to chance (*p* < 0.0001). There was a drop in prevalence between T1 and T2, although this was not statistically significant (*p* = 0.54).

#### 3.2.2. Wellbeing

The reference group for rates of poor mental wellbeing comes from previous research indicating levels of approximately 19% [47]. A two-sample test of proportions indicated that the larger proportion of poor mental wellbeing in the present sample was significantly different to the reference sample at T1 (40.3%, *p* < 0.0001) and T2 (37.2%, *p* < 0.0001). The reduction in poor mental wellbeing between T1 and T2 was not a significant one (*p* = 0.45).

#### 3.2.3. Sleep Disturbance

The proportion of young people reporting sleep disturbance was significantly higher than the reference sample [50] at T1 (30.00%, *p* < 0.0001) and T2 (21.86%, *p* < 0.0001). The decrease in sleep disturbance from T1 to T2 was statistically significant (*p* = 0.03).

#### 3.2.4. Rumination

In the present sample, the proportion of the sample reporting high levels of rumination was significantly higher than estimations derived from previous research (~15%) at T1 (36.12%, *p* < 0.0001) and T2 (29.29%, *p* < 0.0001). The decrease in high levels of rumination from T1 to T2 did not reach statistical significance (*p* = 0.09).

#### 3.2.5. Stress

The levels of stress were significantly higher than previous estimates (25%) at T1 (76.6%, *p* < 0.0001) and T2 (84.0%, *p* < 0.0001). The increase in stress from T1 to T2 was statistically significant (*p* = 0.027).

Given that we measured five facets of mental wellbeing at two timepoints, participants could have reported levels of mental wellbeing that potentially warrant clinical attention a maximum of 10 times over the course of the study. We found that, out of a possible 10 mental wellbeing concerns across T1 and T2 that may have warranted clinical attention, the vast majority of participants had at least one, at either T1 or T2 (97.43%). Throughout the study, we found that 87.15% of participants had two or more mental wellbeing concerns warranting clinical attention, 54.50% had four or more, 25.96% had six or more and 5.92% had eight or more.

### 3.3. Cross-Sectional and Prospective Associations between Lockdown Restriction Severity and Mental Wellbeing

There were no significant differences between baseline characteristics and dropout status apart from gender, where there were marginally more females in the dropout group (see Table 1). The simple differences between low and high lockdown severity groups on the five measures of mental wellbeing are shown in Table 2. In the adjusted SEM models, there was an overall pattern in the data showing that higher verses lower restrictions had a negative effect on the mental wellbeing of young people. At T1, higher restrictions were associated with significantly more depression (B = 2.06, SE = 0.61, *p* = 0.001), more stress (B = 0.93, SE = 0.28, *p* = 0.001) and more rumination (0.93, SE = 0.41, *p* = 0.023). There was a pattern of less wellbeing (B = −2.08, SE = 0.45, *p* < 0.001) and more sleep disturbance (1.02, SE = 0.53, *p* = 0.054) being associated with high restrictions, although the latter was not statistically significant.

The negative effect of higher restrictions persisted prospectively at T2 for depression (B = 2.01, SE = 0. 70, *p* = 0.004), stress (B = 0.99, SE = 0.37, *p* = 0.007), and rumination (1.23, SE = 0.48, *p* = 0.008), and was also significant for wellbeing (B = −2.31, SE = 0.58, *p* < 0.0001) and sleep (1.57, SE = 0.60, *p* = 0.009). Out of a possible 10 mental wellbeing concerns across T1 and T2 that may have warranted clinical attention, the vast majority of participants had at least one at either T1 or T2 (97.43%), 87.15% had two or more, 54.50% had four or more, 25.96% had six or more and 5.92% had eight or more.

### 3.4. Effect of Age and Sex

There were no effects of age at T1 or T2 (all ps > 0.05) and females reported more depression (T1: B = 1.78, SE = 0.72, *p* = 0.014; T2: B = 1.69, SE = 0.89, *p* = 0.056), less wellbeing (T1: B = −2.87, SE = 0.54, *p* < 0.001; T2: B = −1.75, SE = 0.73, *p* = 0.016), more stress (T1: B = 1.50, SE = 0.34, *p* < 0.001; T2: B = 1.73, SE = 0.46, *p* < 0.001), more rumination (T1: B = 1.92, SE = 0.48, *p* < 0.001; T2: B = 1.45, SE = 0.60, *p* = 0.015) and worse sleep (T1: B = 1.18, SE = 0.62, *p* = 0.05; T2: B = 1.75, SE = 0.75, *p* = 0.019).

## 4. Discussion

The results of the current study show a mixed experience of UK university students during the lockdown implemented in response to the COVID-19 pandemic. Overall, students reported poorer health behaviours and lower mental wellbeing indexed by more depression, stress, rumination, sleep disturbance and lower wellbeing—a finding consistent with previous research [8,9,35]. We did not find strong evidence for gender differences on the retrospective measures. However, we did find the well-described gender difference on the majority of indices of mental wellbeing. That is, females reported worse mental wellbeing at both T1 and T2, consistent with previous research [7].

Noteworthy, was the very high prevalence of mental health difficulties that may require clinical attention some nine months into the pandemic lockdown in the UK. In addition, there were some mixed signs that mental wellbeing may be improving from T1 to T2, although the majority of comparisons were not statistically significant. Sleep disturbance significantly improved after a month. It may be that sleep disturbance is less affected than other variables and could therefore be more easily corrected. One longitudinal study, for example, found that whereas depression worsened during the pandemic, sleep quality was not affected [9]. Future research should continue to test this hypothesis which may inform insomnia interventions that can both relieve sleep disturbance and mental health difficulties [57]. Conversely in the present study, stress significantly worsened over the same period. It may be the case that for this student population, already suffering from academic stress [58] and concerns over future prospects, including employment, that symptoms of stress were particularly salient. Stress may have increased because both their academic courses and the COVID-19 lockdowns continued. These two factors could have been sources of stress for many students. Further work should test this hypothesis.

Although low levels of mental wellbeing were detected on all measures in the present sample, our primary outcome was depression. There was an approximate eightfold increase in rates of probable clinical depression, a relative increase similar to that found in previous research [35]. Global depression prevalence estimates in the pandemic have been estimated at ~25–30% [7,15,17,59], a large increase from previous estimates in Europe [60] and worldwide [61]. Our prevalence estimate of probable depression was even higher (~50%), a finding that may be explained by the younger student sample [16]. For instance, Liu et al., reported a depression prevalence of 59% using the PHQ-9 in a student sample [62]. Nevertheless, rates reported typically have been lower than this. For example, two recent studies in university students found depression prevalence estimates on the PHQ-9 of 37% [63] and 35.9% [64]. There was a similar pattern in the data for other measures of mental wellbeing. For example, sleep problems were elevated in the current sample (30% met criteria for sleep disturbance at T1 and approximately 20% at T2), a finding which is consistent with previous research. Insomnia rates during the pandemic have been reported at somewhere between 20% [65,66] and 37% [67].

We also found evidence to suggest not only that mental wellbeing was worse during the pandemic than beforehand, but that mental wellbeing may have been worse for those in higher levels of lockdown restrictions. This finding is consistent with a recent review of 17 systematic reviews that concludes, “the evidence shows the overall impact of COVID-19 restrictions on the mental health and well-being of children and adolescents is likely to be severe.” [68]. Our findings suggest that this conclusion can be extended to young adults at university.

This highlights the need to carefully assess the relative harms caused by lockdowns and to continue to carry out cost benefit analyses. For example, in a UK cost-benefit report, the authors conclude that, “we find that the costs of lockdown in the UK are so high relative to likely benefits that continuing the lockdown for three months was unlikely to be warranted.” [69]. Similarly, a review of the literature in Canada suggests that assumptions made early on in the pandemic may have been unrealistic, for example assuming that lives lost would be independent of age [70]. This has meant that young people suffered unnecessarily at the hands of a number of potent iatrogenic effects of “lockdown” [71]. That is to say, our response to the COVID-19 pandemic may have unwittingly unleashed a litany of adverse sequelae including but not limited to mental health problems [72].

We have some sympathy with the view expressed elsewhere that lockdowns were, perhaps initially, both “desperate” and “defendable choices” [73], but now that so much is known on the resultant harms, we believe lockdowns should be avoided in the future. There is, for example, little evidence of benefit to the more restrictive NPIs over less restrictive ones [74] and more broadly, a recent analysis suggests that while long periods of lockdown do not reduce the fatality rate, they do have a negative impact on economic growth [75]. It should also be considered that levels of infection from Sars-Cov-2 may have been declining before lockdowns were introduced in the UK [76].

It is important to also stress that NPIs, such as lockdown, may have unintended pleiotropic adverse effects on young people, unmeasured in the current study, that may indirectly lead to mental disorder [77]. These may include loneliness, isolation [34] and exposure to domestic violence [78,79].

Perhaps most concerning, however, is the potential for lockdown to cause hysteresis effects for young people at what is a sensitive developmental period. That is, scarring [80] that is likely to cause long-term damage and adversely affect future prospects in terms of health and wellbeing, education and quality of life. The crux of this problem lies in both the singular potency and the interplay between two factors, which in combination, have the potential to cause long-term harm. These are: the prolonged disruption of education and the unfolding of common mental health disorders. Both the lack of education and the emergence of mental health problems are heavyweights in their own right when it comes to foreshadowing negative outcomes. Unfortunately, however, their combined adverse effect is likely to be severe because the two are inextricably linked. Only longitudinal research will be able to assess this hypothesis.

In their review of the evidence on child and mental health in the pandemic, Heneghan et al. [68] highlight that socialisation was a key protective factor, as were prosocial behaviours (e.g., helping others). Thus, it will be important to ensure that opportunities for students to socialise face-to-face are maintained and prioritised as the world begins to recover from the pandemic. Universities may also consider increasing their offer of supporting community voluntary work to boost opportunities for prosocial behaviours. There is an increased need for mental health support at universities which will require significant investment. More broadly, long-term planning and investment in mental health is needed post-pandemic, particularly for children and young people, a group that has been disproportionately affected [81]. One aspiration is that strategies put in place now, such as strengthening mental health systems and investing in health workers, will not only help to mitigate some of the harms caused by the current lockdowns but also serve to increase resilience in mental health systems in the future [82].

## 5. Strengths and Limitations

The present study has a number of strengths. First, we were able to capture measurements on the prevalence of five important indices of mental wellbeing in young people: depression, depressogenic cognition (rumination), wellbeing and sleep disturbance. In this way we are able to highlight the potential clinical need in this vulnerable group. Second, we designed the study to provide important information on mental wellbeing during the COVID-19 pandemic. Third, we provide much-needed longitudinal data on the effects of lockdown on young people. Fourth, the study is one of the few to date that has measured the dose-response effect of lockdown severity on mental wellbeing. Fifth, the study was well-powered and included robust multivariate analyses. Although there was some attrition in the study at the one-month follow up, we did not find any differences on baseline measures between those who did and did not drop out from the study. In addition, we used a maximum likelihood estimator in the inferential analysis to account for missing data.

Although a strength of the study was the timing of the baseline assessment (during the COVID-19 pandemic) and the inclusion of a month follow up, the study follow up is necessarily limited in time (though shorter follow-up periods have been called for in mental health research where longer follow-up periods do not increase predictive utility of mental health outcomes [83]). Although beyond the scope of the present undertaking, future studies should include longer-term follow up periods. Another limitation to the present study was that the follow up period encompassed the winter break for the majority of students, where it is likely that many engaged in some restful activities, and many would receive respite and social support from family and friends. In this case, it is entirely possible that the negative effects of lockdown could have been mitigated by social support (as might be expected from the work of Heneghan et al. noted above) [68]. Therefore, all things being equal we would expect the potential clinical need to increase in the absence of said social support. However, it is important to note that all students were preparing for examinations over this period, scheduled for January 2021, which would be a source of stress. Clearly, future studies that include longer follow up periods should include relevant measures, including social support, loneliness and isolation. Finally, while a strength of this study was the quantitative approach that included well-validated psychometrically sound measures of mental wellbeing, it is limited in that it was unable to explore the lived experience of young people through the lockdowns. Future qualitative studies could help us to further understand the detailed experiences and trajectory of students through the COVID-19 lockdowns.

## 6. Conclusions

In summary, young people were left relatively unscathed by Sars-CoV-2 during the COVID-19 pandemic and yet the evidence presented here suggests that lockdowns are likely causing psychopathology that may in turn herald a chronic course of negative outcomes, including educational, social, economic, physical, societal and personal costs. Given that loss of education and poor mental health are tightly intertwined, we encourage policy makers to think again when considering this strategy in the future in the hope to avoid further long-term damage to children and young people. We also hope that students and young people will be given the opportunity to repair through socialisation processes as well as mental health interventions. The dramatic increase in mental health difficulties and potential clinical need highlighted in the present study and elsewhere requires an increase in mental health support at universities and in the wider community. As a society we must be prepared to make significant investment to fund such initiatives.

## Figures and Tables

**Figure 1 ijerph-19-01132-f001:**
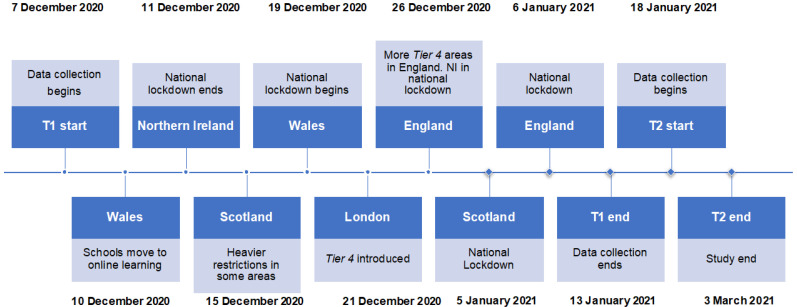
Study timeline.

**Figure 2 ijerph-19-01132-f002:**
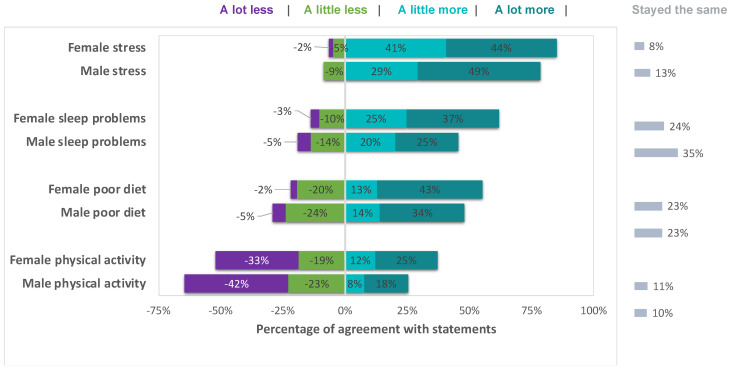
Change in mental wellbeing and health behaviour. Percentages represent participants’ agreement with question statements (see Section 2.3.1 for statement wording).

**Table 1 ijerph-19-01132-t001:** Baseline characteristics of participants with and without missing data at T2.

	Participants without Missing Data (*n* = 254)	Participants with Missing Data (*n* = 135)	*p*-Value ^a^
Demographics			
Age (mean, sd)	21.12 (1.60)	20.88 (1.65)	0.17
Gender (%, female)	73.08	81.60	0.05
Year of study (%)			0.18
Year 1	22.98	25.78	
Year 2	20.97	30.47	
Year 3	35.48	28.91	
Year 4	8.87	5.47	
Postgraduate	11.69	9.38	
PHQ (mead, sd)	11.12 (5.42)	10.29 (6.06)	0.39
WELL	17.06 (7.94)	17.69 (6.84)	0.42
PSS	9.41 (2.62)	9.66 (2.96)	0.34
RUM	11.29 (3.64)	11.09 (3.94)	0.63
JSS	8.29 (5.23)	8.91 (4.73)	0.60

Note. ^a^ Multivariate Analysis of Variance (MANOVA) or chi-square tests.

**Table 2 ijerph-19-01132-t002:** Means and standard deviations for the five wellbeing measures at T1 and T2, by lockdown restriction severity groups.

Variables	Lower Restrictions (Mean, SD)	Higher Restrictions (Mean, SD)	*p*-Value for Difference Test (One-Way ANOVA)
PHQ			
T1	10.14 (5.46)	11.88 (5.73)	0.007
T2	9.31 (5.71)	11.15 (5.24)	0.014
WMWBS			
T1	20.35 (3.97)	18.35 (4.47)	<0.001
T2	21.18 (4.58)	18.96 (4.50)	<0.001
PSS			
T1	9.21 (2.86)	10.05 (2.51)	0.006
T2	10.16 (3.03)	11.07 (2.72)	0.019
RUM			
T1	10.92 (3.62)	11.80 (3.96)	0.046
T2	10.36 (3.74)	11.55 (3.56)	0.017
JSS			
T1	8.51 (4.86)	9.18 (4.83)	0.210
T2	7.56 (4.46)	9.18 (4.89)	0.009

Note. PHQ = patient Health Questionnaire-8; WMWBS = The Warwick-Edinburgh Mental Wellbeing Scale; PSS = Perceived Stress Scale-4; RUM = The Brooding Scale; JSS = The Jenkins Sleep Scale.

## Data Availability

Anonymised data may be made available upon request.

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
