# Peer review of "Mental Health and Wellbeing in Young People in the UK during Lockdown (COVID-19)"

_ijerph, 2022, doi:10.3390/ijerph19031132_

Round 1
Reviewer 1 Report
In this manuscript, the authors report on an empirical investigation of the mental wellbeing and potential clinical need in UK university students during the COVID 19 pandemic and examine wellbeing in relation to lockdown restriction severity. Strengths of the analysis are a decent sample size (n = 389) and longitudinal (shortitudinal) data. Therefore, this manuscript contributes substantially to the existing literature by highlighting the potential clinical need in this vulnerable group.
Throughout, I would encourage the authors to consider whether they really focused on ‘wellbeing’ or depressive symptoms? I think “wellbeing” is perhaps a bit misleading, given that the PHQ-8 is really a measure of the latter, and was their primary outcome measure.
Introduction
Sets the context nicely, explaining the important concepts and timeline, and giving relevant literature to funnel towards the aim and research questions of this study specifically. I commend the authors for their consideration of the developmentally sensitive period of 18-25 year olds, and I wonder if one or two more sentences could be added to say how lockdown measures may have impacted specifically on this age group.
Methods
Well described. Particularly impressive description of measures which is neat but comprehensive. What feels like it is need is a procedure section – how were the participants recruited? Where was the study advertised? How did they complete the measures (online?)? How many times were they reminded about follow up measures? Any incentives offered for study participation? Ethics?
Results
Well reported with good use of tables and figures. Minor point – check the first 2 sentences of this section which seem to be instructions to the authors rather than part of the write up?
Discussion
Generally very good with relevant possible explanations. I’d like to see more explicit consideration of the strengths and limitations of this study, and also to see links made to qualitative studies which might help us to understand more the journey of UK students through the pandemic.
Minor point – check the final 2 sentences of this section which seem to be instructions to the authors rather than part of the write up?
Author Response
Please see attachment. All responses to all reviewers are included.

Reviewer 2 Report
Dear Authors,
I found your work very interesting, considering the scientific soundness of this research. Moreover, the topic of the impact of Covid-19's on Young needs to be improved in the current literature, so I am glad to review this work.
However, I found different facets that need to be improved and I am particularly concerned about the time-lapse you chose. Here there are my comments:
The current Study
- The timeline of this study needs to be specified, in light of the short-longitudinal study. It is necessary to specify the specific date of T1 (start-end) and T2 (start-end).
- The third hypothesis has different weaknesses: if in T2 there is a general lockdown, supposing a heavy level of restriction, why you did not suppose a decrease in mental health? Moreover, the time between T1 and T2 is very short to take a possible consideration of a persistent impact on the time. Another important point, that should be taken into consideration in the limits, is the Christmas holiday. They could influence the final data as they are an important moment for each individual (increasing or decreasing the stress levels). This is a methodological and clinical aspect that need to be taken seriously, considering also the short time between T1-T2. I suggest revisiting this hypothesis.
Materials and Method
- There is a typo in the reference in line 185, it is not in number.
- In the section of Lockdown Status, it is not clear the criteria you adopted to determine the severity level for each status of restriction. I please the author to specify this point, maybe with a table
- It is missing a Procedure section. There is no information about the recruiting method of participants, the modality taken to present questionnaires (online?), the time needed to complete questionnaires (it could be important information to understand the huge drop out of the present study)...
- In the paper, the authors talk about the "10 mental wellbeing concerns". They are not clear and, if coherent, they need to be added in this section
Results
- Lines 255-257 seem to be a typo of the editing docx.
- The number of drop-outs between T1-T2 could be explored taking in consideration the level of severity of restrictions in T1
Discussion
- In this section, the Authors did not discuss some interesting data (the gender difference; the increase of sleep quality between T1-T2). I think that some considerations can be taken, also exploring the existing literature.
- Lines 434-437 seem to be a typo
Conclusions
In this section, the authors take excessive conclusions, with ethical problems, as they suggest to avoid the strategy of lockdowns in the future. There are no sufficient data and facets taken into consideration to make a similar statement. It seems more appropriate a proposal based on the invitation to police makers to take into account the impact of this kind of restriction on mental health, building, in parallel with these decisions, specific programs of psychological support for citizens and especially for the younger.
Limits
There is no limit section, that must be added.

Author Response

(The authors gave the same response as above.)

Reviewer 3 Report
Figure 2 is not very informative. in 3.1 of results section, the authors metion several percentage values in manuscript (in line 260 to 264), but figure 2 stresses different percentage numbers in the bar. Authors should change the format to match what authors want to stress in the figure with what in manuscript. The other inforamtion in the figure should be mentioned at least once in the legend of each figure. For example, what is 'agreement with statement'? I could not see any explanation in manuscript.
Description of statistic analyses should be done more precisely in this study.
For example, in line 266, I cannot see which statistic method the authors used. In table 2, they said ANOVA is used for p-value for difference test, but ANOVA should be used more than three groups. Is this anova one way or two way ? repeated measure? They don't give any information about detail on statistics.
Author Response

(The authors gave the same response as above.)

Round 2
Reviewer 2 Report
You did a great job
Author Response
Thank you.